# Chest CT Total Severity Score on Admission to Predict In-Hospital Mortality in COVID-19 Patients with Acute and Chronic Renal Impairment

**DOI:** 10.3390/diagnostics12071529

**Published:** 2022-06-23

**Authors:** Samar Tharwat, Gehad A. Saleh, Marwa Saleh, Ahmad M. Mounir, Dina G. Abdelzaher, Ahmed M Salah, Mohammed Kamal Nassar

**Affiliations:** 1Rheumatology & Immunology Unit, Department of Internal Medicine, Faculty of Medicine, Mansoura University, Mansoura 35516, Egypt; 2Diagnostic Radiology Department, Faculty of Medicine, Mansoura University, Mansoura 35516, Egypt or gehadsaleh@mans.edu.eg (G.A.S.); ahmedmounir@mans.edu.eg (A.M.M.); gamaldina2012@gmail.com (D.G.A.); 3Mansoura Nephrology & Dialysis Unit (MNDU), Department of Internal Medicine, Faculty of Medicine, Mansoura University, Mansoura 35516, Egypt; marwasaleh@mans.edu.eg (M.S.); m_kamal@mans.edu.eg (M.K.N.); 4Nephrology Unit, Department of Internal Medicine, Faculty of Medicine, Zagazig University, Zagazig 44519, Egypt; ahmedsalah_400@yahoo.com

**Keywords:** COVID-19, chest CT total severity score, acute renal impairment, chronic kidney disease

## Abstract

**Aim:** To identify the predictors of in-hospital mortality in patients with coronavirus disease of 2019 (COVID-19) and acute renal impairment (ARI) or chronic kidney disease (CKD), and to evaluate the performance and inter-reader concordance of chest CT total severity scores (TSSs). **Methods:** This retrospective single-center study was conducted on symptomatic COVID-19 patients with renal impairment (either acute or chronic) and a serum creatinine of >2 mg/dL at the time of admission. The patients’ demographic characteristics, clinical data, and laboratory data were extracted from the clinical computerized medical records. All chest CT images obtained at the time of hospital admission were analyzed. Two radiologists independently assessed the pulmonary abnormalities and scored the severity using CT chest total severity score (TSS). Univariate logistic regression analysis was used to determine factors associated with in-hospital mortality. A receiver operating characteristic (ROC) curve analysis was performed for the TSS in order to identify the cut-off point that predicts mortality. Bland–Altman plots were used to evaluate agreement between the two radiologists assessing TSS. **Results:** A total of 100 patients were included, with a mean age of 60 years, 54 were males, 53 had ARI, and 47 had CKD. In terms of in-hospital mortality, 60 patients were classified in the non-survivor group and 40 were classified in the survivor group. The mortality rate was higher for those with ARI compared to those with CKD (*p* = 0.033). The univariate regression analysis showed an increasing odds of in-hospital mortality associated with higher respiratory rate (OR 1.149, 95% CI 1.057–1.248, *p* = 0.001), total bilirubin (OR 2.532, 95% CI 1.099–5.836, *p* = 0.029), lactate dehydrogenase (LDH) (OR 1.001, 95% CI 1.000–1.003, *p* = 0.018), CRP (OR 1.010, 95% CI 1.002–1.017, *p* = 0.012), invasive mechanical ventilation (MV) (OR 7.667, 95% CI 2.118–27.755, *p* = 0.002), a predominant pattern of pulmonary consolidation (OR 21.714, 95% CI 4.799–98.261, *p* < 0.001), and high TSS (OR 2.082, 95% CI 1.579–2.745, *p* < 0.001). The optimum cut-off value of TSS used to predict in-hospital mortality was 8.5 with a sensitivity of 86.7% and a specificity of 87.5%. There was excellent interobserver agreement (ICC > 0.9) between the two independent radiologists in their quantitative assessment of pulmonary changes using TSS. **Conclusions:** In-hospital mortality is high in COVID-19 patients with ARI/CKD, especially for those with ARI. High serum bilirubin, a predominant pattern of pulmonary consolidation, and TSS are the most significant predictors of mortality in these patients. Patients with a higher TSS may require more intensive hospital care. TSS is a reliable and helpful auxiliary tool for risk stratification among COVID-19 patients with ARI/CKD.

## 1. Introduction

The novel coronavirus disease (COVID-19), caused by the severe acute respiratory syndrome coronavirus 2 (SARS-CoV-2), first appeared in Wuhan, China in December 2019 as an unknown entity before being identified on 7 January 2020 [1]. The COVID-19 outbreak has dramatically impacted the activities of health care systems [2]. Clinical presentation is diverse, ranging from asymptomatic to severe disease. Individuals may develop COVID-19, which is an influenza-like illness that primarily affects the respiratory system. Major symptoms include fever, cough, and dyspnea [3], while minor symptoms include altered sense of smell and taste [4], gastrointestinal complaints [5], and skin manifestations [6].

Acute respiratory distress syndrome (ARDS) occurs in some patients and necessitates intubation and mechanical ventilation (MV). Multi-organ dysfunction is common in these patients and is associated with a significant risk of death [7]. The early detection of COVID-19 in patients with a high risk of death can help to improve patient management and enables the more effective use of resources throughout the healthcare system [8].

Age is the most important factor for mortality; older patients have a higher risk of mortality, as evidenced by retrospective patient analyses [9] and subsequent public health guidelines [10]. Patients with low oxygen saturation have a higher risk of death, which supports results that link hypoxemia to mortality [11] and the documented incidence of shortness of breath in patients with a severe risk of mortality [12]. Respiratory failure has been identified as one of the leading causes of COVID-19 mortality [13]. Other chronic diseases, such as arterial hypertension, ischemic heart disease, atrial fibrillation, and cancer, are linked to a high mortality rate [14]. Furthermore, chronic kidney disease (CKD) has recently been identified as a significant risk factor for COVID-19 mortality [15,16]. Additionally, COVID-19 has a high rate of renal involvement (4–37%) [17]. Acute kidney injury was reported in 5–15% of COVID-19-infected patients.

The CT findings of COVID-19 infection are similar to those described in viral pneumonias [18], with multifocal ground-glass opacities and peripheral consolidation [19,20]. Although these findings lack the imaging specificity necessary to diagnose COVID-19, CT could be employed to offer an objective assessment of the extent of the disease. A variety of chest CT grading systems have been proposed to assess the severity of pulmonary affection in COVID-19 infection. A recent study examined the diagnostic performance of five different CT chest severity scoring systems and found that CT total severity score (TSS) had the highest specificity and utilized the least amount of time when compared to other scoring systems [21].

However, studies on COVID-19 infection and CT results in patients with renal impairment are limited. To the best of our knowledge, no available published studies have investigated the factors associated with mortality and whether TSS can predict in-hospital mortality in patients with renal impairment.

Therefore, the aim of this study was to explore the clinical characteristics, laboratory characteristics, and chest CT findings in non-surviving COVID-19 patients with ARI/CKD and to evaluate the diagnostic performance and inter-reader concordance of TSS for predicting in-hospital mortality among these patients.

## 2. Materials and Methods

### 2.1. Patients

This is a retrospective single-center study conducted on symptomatic COVID-19 patients who were admitted to Mansoura University hospital between June 2020 and May 2021. The inclusion criteria included the following: (a) renal impairment (either acute or chronic) with serum creatinine >2 mg/dL at the time of admission, (b) aged over 18 years, and (c) those who underwent chest CT at the time of hospital admission. Any patients with other associated infections, lung malignancy, tuberculosis, or other chronic chest problems were excluded from the start. The study was approved by the Institutional review board of Mansoura University (Approval No: R.21.11.1530) and a waiver of consent to review medical records was received.

One hundred and fifty consecutive patients fulfilled the inclusion criteria and were initially enrolled in this study. Of them, 109 patients were confirmed by a real-time reverse transcription polymerase chain reaction (RT-PCR) throat swab [22]. Then, 9 patients were excluded: 2 patients with negative findings at chest CT and 7 patients with incomplete clinical or laboratory data. Finally,100 patients were included in the study. A flow chart illustrating participant selection in the study is demonstrated in Figure 1.

### 2.2. Clinical and Laboratory Data

The patients’ demographic, clinical, and laboratory data were extracted from the clinical computerized medical records obtained from Mansoura University hospital. The demographic data included age and sex. The clinical data included the type of renal impairment (either acute or chronic), associated comorbidities, presenting symptoms of COVID -19 infection, and vital data at the time of hospital admission. The laboratory data mainly included complete blood count (CBC), serum creatinine (Cr), liver function tests (LFTs), lactate dehydrogenase (LDH), erythrocyte sedimentation rate (ESR), D dimer, C-reactive protein (CRP), and arterial blood gases. All other data regarding the total duration of hospital stay, need for intensive care unit (ICU) admission, oxygen supply, or invasive MV were also obtained.

### 2.3. Chest CT Acquisition

All patients underwent chest CT imaging without a contrast agent at time of admission on a 16-detector CT scanner (Bright speed; GE healthcare). All of the patients were examined in a supine position, with images acquired during a single inspiratory breath-hold. The scanning range was from the base of the neck to the level of the upper pole of the kidneys. CT scan parameters were as follows: X-ray tube parameters, 120 KVp, 350mAs; rotation time, 0.5 s; pitch, 1.0; section thickness, 5 mm; intersection space, 5 mm; and additional reconstruction with a slice thickness of 1.5 mm. Scans were reviewed at a window width and level of 1000 to 2000 HU and −700 to −500 HU, respectively, in order to assess the lung parenchyma. Images were reconstructed in axial, coronal, and sagittal reformats.

### 2.4. Qualitative CT Image Analysis

All chest CT images obtained at the time of hospital admission were analyzed by one reviewer with more than 11 years of experience in thoracic imaging. The main findings from the chest CT scans were described as follows: ground-glass opacities (GGOs), pulmonary consolidation, and crazy paving pattern based on the standard glossary for thoracic imaging reported by the Fleischner Society [23], and the predominant pattern and its laterality were determined. The chest CT scans were also evaluated for the presence of pleural effusion, pulmonary nodules, bronchial dilatation, mediastinal lymphadenopathy, and pericardial effusion.

### 2.5. Quantitative CT Image Analysis

To evaluate the severity of pulmonary parenchymal involvement, the extent of the abnormalities was quantified by the total severity score (TSS). TSS is mainly a quantitative score used to assess inflammatory abnormalities in each of the five lobes of both lungs, including the existence of GGOs, consolidation, or mixed GGO [24]. All CT images were independently analyzed by two chest radiologists in consensus (GAS and AMM with 11 and 10 years of experience in chest CT scan interpretation, respectively). Both radiologists were blinded to both clinical and laboratory data. The TSS was calculated for each of the 5 lobes in all patients. According to the extent of pulmonary involvement, each lobe could be scored from 0 to 4 points as the following: 0, no involvement; 1, from 1 to 25% involvement; 2, from 26 to 50% involvement; 3, from 51 to 75% involvement; and 4, more than 75% involvement. The sum of each individual lobar score resulted in the TSS, which ranged from 0 to 20.

## 3. Statistical Analysis

Data were collected, revised, and analyzed using IBM SPSS Statistics version 21.0 for Windows (IBM Corp., Armonk, NY, USA) on a personal computer. All quantitative values were described using medians (minimum-maximum) or means ± standard deviation (SD), while qualitative variables were described using the numbers and percentages (%) of cases. Patients were assigned to two groups according to in-hospital mortality (non-survivors and survivors). The normal distribution of continuous variables was examined using the Shapiro–Wilk test. Then, the significance of differences between the two groups was determined using the independent samples *t* test for normally distributed variables or the Mann–Whitney test for non-parametric variables, as appropriate. For comparisons between qualitative variables, Chi-squared or Fisher’s exact tests were utilized, as appropriate.

Univariate and multivariate logistic regression analyses were used to determine factors associated with in-hospital mortality. Subsequently, a receiver operating characteristic (ROC) curve analysis was performed for the TSS in order to identify the cut-off point above which morality is likely, considering the readings of the first radiologist. Bland–Altman plots were used to evaluate agreement between the two radiologists in their assessment of the TSS.

## 4. Results

### 4.1. Demographic, Clinical and Laboratory Characteristics

A total of 100 patients hospitalized with PCR-confirmed COVID-19 and associated renal impairment were included in this retrospective study. The mean age was 60 ± 15 years, and 54 patients were males, 53 had ARI, 47 had CKD, 23 patients were hypertensive, and 23 were diabetic. The typical presenting symptoms of COVID-19 infection were dyspnoea (86), fever (75) and cough (65), and acute confusional state (6).

According to in-hospital mortality, 60 patients were classified in the non-survivor group and 40 in the survivor group. The characteristics of non-surviving and surviving patients are illustrated in Table 1. The non-survivor and survivor groups had no statistically significant differences regarding age (*p* = 0.261) and gender (*p* = 0.060). Non-survivors had a significantly higher association with ARI than CKD (61.7 vs. 40%, *p* = 0.033), liver cirrhosis (15% vs. 2.5%, *p* = 0.047), higher body temperature (38.56 vs. 38.01, *p* = 0.003), and respiratory rate (<0.001).

By comparing the laboratory data between non-survivors and survivors at the time of admission, we found that non-survivors had higher serum levels of ALT (*p* = 0.030), AST (*p* = 0.007), bilirubin (*p* = 0.012), LDH (*p* = 0.001), and CRP (*p* = 0.003), and lower levels of serum albumin (*p* = 0.010) and arterial oxygen saturation (*p* = 0.001).

Although there was no significant difference between the two groups regarding the total duration of hospital stay, ICU admission was significantly higher in the non-survivor group (95% vs. 65%, <0.001).

### 4.2. CT Chest Features

GGOs were present in all patients (100%). Pulmonary consolidation was observed in 71 patients and was more prevalent in non-survivors (90% vs. 42.5%, *p* < 0.001). Crazy paving pattern was found in 17 patients and all of them were in the non-survivor group. Demonstrative non-severe (survivor) and severe (non-survivor) COVID-19 cases are presented in Figure 2 and Figure 3. Other chest CT findings are illustrated in Table 2. The predominant pattern of abnormality upon hospital admission was GGO (64), followed by pulmonary consolidation (34) and the crazy paving pattern (2).

### 4.3. Determinants of Mortality

The univariate logistic regression analysis showed that respiratory rate (OR 1.149; [95% CI, 1.057–1.248], *p* = 0.001), bilirubin (OR 2.532; [95% CI,—1.099–5.836], *p* = 0.029), LDH (OR 1.001; [95% CI, 1.000–1.003], *p* = 0.018), CRP (OR 1.010; [95% CI, 1.002–1.017], *p* = 0.012), the need for invasive MV (OR 7.667; [95% CI, 2.118–27.755], *p* = 0.002), the presence of pulmonary consolidation (OR 0.082; [95% CI, 0.029–0.235], *p* < 0.001), and higher TSS (OR 2.082; [95% CI1.579–2.745], *p* < 0.001) were the main risk factors for mortality among COVID-19 patients with ARI/CKD, as shown in Table 3.

Among the variables studied, CRP, ICU admission, and TSS were significant predictors of mortality when applying multivariate logistic regression, as illustrated in Table 4.

### 4.4. TSS to Predict Mortality

The ROC curve analysis revealed that TSS was statistically significantly higher in non-survivors (*p* < 0.001) (Figure 4). The area under the curve (AUC) for discriminating between non-survivors and survivors was excellent: 0.945 (CI, 0.901–0.989). The optimum cut-off value of TSS used to predict in-hospital mortality was 8.5, with a sensitivity of 86.7% and a specificity of 87.5%.

### 4.5. Interobserver Agreement

Two-hundred observations were described in order to calculate the TSS. The interclass correlation (ICC) for TSS between the two observers was 0.956. The Bland–Altman plot (Figure 5) revealed excellent interobserver agreement between the two independent observers in quantitative lung assessment using the TSS.

## 5. Discussion

To the best of our knowledge, this is the first study in Egypt that analyzes in-hospital mortality among COVID-19 patients with ARI/CKD and evaluates the performance and inter-reader concordance of TSS among these patients. In this single-center, observational, retrospective study, we found that the in-hospital mortality among COVID-19 patients with ARI/CKD was very high and was associated with ARI more than CKD. The most predominant CT chest finding was GGO, which was present in almost all patients. The TSS was significantly higher in non-survivors (*p* < 0.001). If the TSS is more than 8.5, it may indicate mortality in these patients with 86.7% sensitivity and 87.5% specificity. Additionally, there was excellent interobserver agreement between the two radiologists in quantitative pulmonary assessment using the TSS.

In this study, we observed a high rate of in-hospital mortality among our cohort. This is in agreement with the findings of another study in which Gasparini and coauthors [25] looked at the link between acute and chronic renal disease and clinical outcomes in 372 patients with COVID-19 who were admitted to four regional intensive care units. In-hospital mortality was higher in patients with ARI/CKD compared to individuals with preserved renal function (50% vs. 21%; *p* = 0.001). Chu and co-authors [26] also found that patients with COVID-19 and ARI had a considerably greater mortality rate than those with COVID-19 but no renal impairment (91.7% vs. 8.8%; *p* < 0.001). In a population-based cohort study from England that was conducted to assess the risk factors associated with mortality among COVID-19 patients with type 1 and type 2 diabetes [27], impaired renal function was also linked to an increase in COVID-19-related mortality among these patients.

Our data demonstrated a higher mortality rate among COVID-19 patients with ARI in comparison to those with CKD (61.7% vs. 40%). This is in agreement with the findings from a study conducted by Zahid [28], which showed that the in-hospital mortality of COVID-19 patients with ARI was 71.1%; in-hospital mortality was linked to ARI that was larger than or equal to stage 2. Therefore, clinicians should seek to detect and treat renal injury in COVID-19 patients before the disease progresses to the point at which mortality rises dramatically.

This study showed that patients with liver cirrhosis had a higher risk of mortality. In a multicenter, concurrently enrolled study in North America [29], age/gender-matched COVID-19 patients with cirrhosis had a greater mortality rate than those with COVID-19 alone. Additionally, the registry data in COVID-19 patients with cirrhosis showed an association with poor prognosis, with a mortality rate approaching 40% [30,31]. The higher mortality among these patients may be related to an altered gut–liver axis and inherent immune dysfunction [29].

In the present study, blood levels of ALT, AST, bilirubin, LDH, and CRP on admission were significantly higher among non-survivors. Moreover, patients with low arterial oxygen saturation and patients who were admitted to the ICU were more likely to die. This is in agreement with published data for all COVID-19 patients. Deranged liver chemistry showed severe COVID-19 and predicted mortality in a metanalysis of 22 studies involving 3256 COVID-19 patients [32]. Furthermore, LDH elevation reflects tissue/cell destruction and is seen as a common indicator of tissue/cell damage, suggesting viral infection or lung injury such as that caused by SARS-CoV-2 pneumonia. LDH has been recognized as a key biomarker for the activity and severity of idiopathic pulmonary fibrosis. LDH levels rise significantly in individuals with severe pulmonary interstitial disease, and it is one of the most important prognostic indicators of lung injury [33,34]. Hence, it is not surprising that LDH was higher in the non-survivor group of our study.

Chest CT has a well-established role in the diagnosis and management of COVID-19 patients during their hospital stay [35]. COVID-19 has a wide range of pulmonary CT characteristics [36]. In this study, all patients had GGOs. This finding is similar to the results of multiple previous studies in which the majority of COVID-19 patients had GGO and mixed GGO with consolidation [21,37,38]. However, pulmonary consolidation was found in 71 individuals, with non-survivors having a higher rate of pulmonary consolidation (90% vs. 42.5%, *p* = 0.001). Our results revealed a statistically significantly higher prevalence of crazy paving pattern in non-survivors compared to survivors, as this pattern was discovered in 17 individuals, all of whom were non-survivors. This prevalence is in agreement with previous literature [21,39,40]. However, there was no statistically significant difference in GGO among survivors and non-survivors, which aligns with the results of a previous study [41]. GGO is a marker of the early and active stage of disease, which can progress and worsen patients’ health. Furthermore, CT scans showing mixed GGO with consolidation, when paired with the pathological abnormalities of COVID-19 such as hyaline membrane development and increased inflammatory exudate of the alveolar space, are linked to increased mortality in these patients [7]. The frequency of crazy-paving pattern in severe cases may indicate a mixture of alveolar enema, bacterial superinfection, and interstitial inflammatory changes [42].

To assess the degree of lung involvement, we utilized a semi-quantitative scoring system previously employed by Li et al. [24]. Our results revealed that TSS was statistically significantly higher among non-survivors compared to survivors. This agreed with the findings of a recent study [21]. Therefore, early ICU admission may be beneficial for patients with a higher TSS. This may help with patients’ disposal in emergency rooms, especially in situations where ICU resources are restricted.

In patients with COVID-19, qualitative and quantitative chest CT parameters are predictors of mortality [43]. In our study, TSS revealed excellent diagnostic accuracy using a cut-off value of 8.5 or above for the detection of in-hospital mortality in COVID-19 patients with ARI/CKD, with 86.7% sensitivity and 87.5% specificity. Similar results were found in a retrospective single study from China conducted on 78 COVID-19 patients investigating the relationship between COVID-19 imaging findings and clinical categorization. The researchers found that a TSS threshold of 7.5 showed a sensitivity of 82.6% and a specificity of 100% for diagnosing severe–critical COVID-19 infection [24]. Another study revealed concordant results, with 77.3% sensitivity and 90.5% specificity using a relatively higher TSS cutoff value of 12 to differentiate severe critical cases from non-severe cases [21].The combination of both qualitative and quantitative assessment could help to distinguish severe critical cases from non-severe cases, and could aid in the rapid diagnosis of and care for critical cases, thereby decreasing mortality and improving patient outcomes.

Additionally, interobserver agreement was determined using Bland–Altman plost which yielded a statistically significant excellent agreement between the two observers in assessing TSS (ICC = 0.956, *p* < 0.001). This was similar to a recent study which also revealed excellent agreement for quantitative lung assessment using TSS and other CT chest scoring systems [21].

There are some limitations to this study. First, it is a single-center retrospective study. Secondly, there was no precise information about the exact duration of time between the commencement of symptoms and CT acquisition. Lastly, no patients underwent a lung biopsy to mimic the histopathological pulmonary changes. Future studies evaluating the performance of artificial intelligence and deep-learning-based tools against a radiologist-based severity scoring system are recommended.

In conclusion, in-hospital mortality is high in COVID-19 patients with ARI/CKD, especially those with ARI. High bilirubin, a predominant pattern of pulmonary consolidation, and TSS are the most significant predictors of mortality in these patients. Patients with higher TSS may require more intensive hospital care. TSS is a reliable and helpful auxiliary tool for risk stratification in COVID-19 patients with ARI/CKD.

## Figures and Tables

**Figure 1 diagnostics-12-01529-f001:**
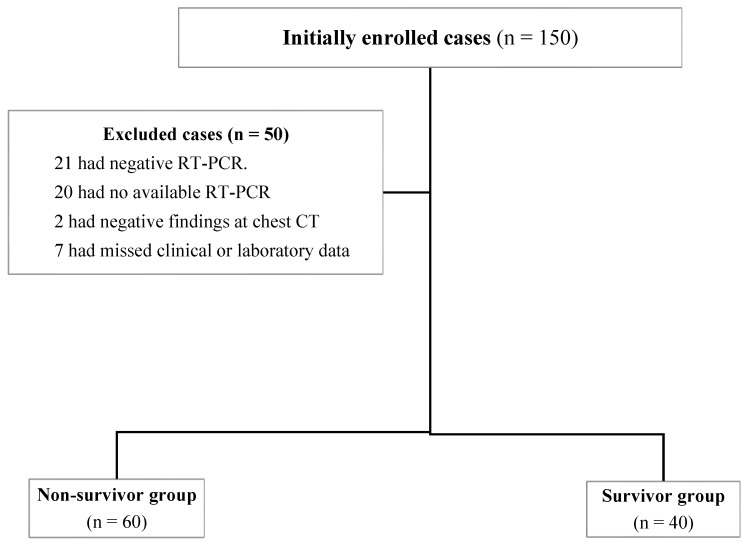
The study flowchart.

**Figure 2 diagnostics-12-01529-f002:**
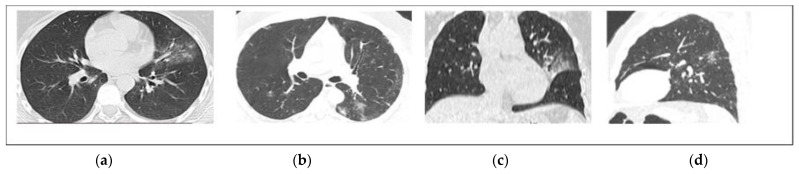
Non-contrast chest CT axial (**a**,**b**), coronal (**c**), and sagittal (**d**) reconstructed images for a 38-year-old male with mild COVID-19 infection. CT images show small ground-glass opacities in both lobes with left upper lobe consolidation. The TSS is 4.

**Figure 3 diagnostics-12-01529-f003:**
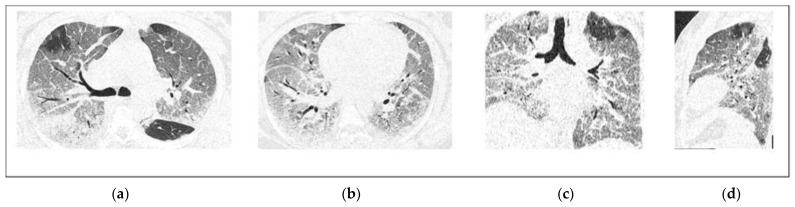
Non-contrast chest CT axial (**a**,**b**), coronal (**c**), and sagittal (**d**) reconstructed images for a 52-year-old female with severe COVID-19 infection. CT images show large patchy ground-glass opacities and a predominant crazy paving pattern in multiple lung segments in both lobes. The TSS is 18.

**Figure 4 diagnostics-12-01529-f004:**
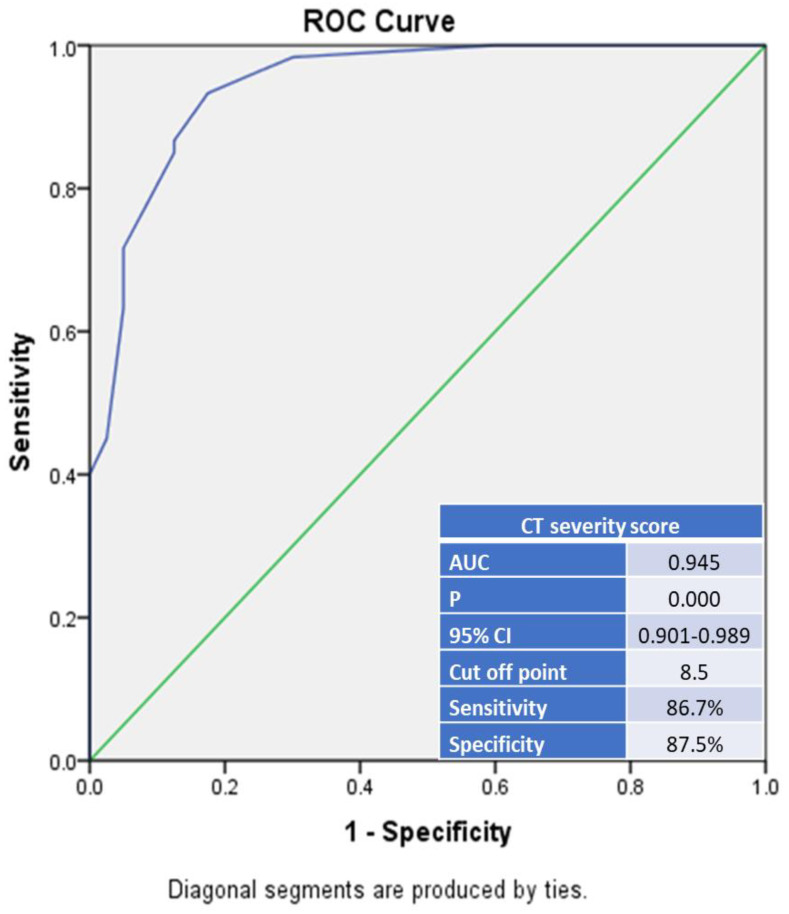
ROC curve for chest CT severity score sensitivity and specificity to predict mortality.

**Figure 5 diagnostics-12-01529-f005:**
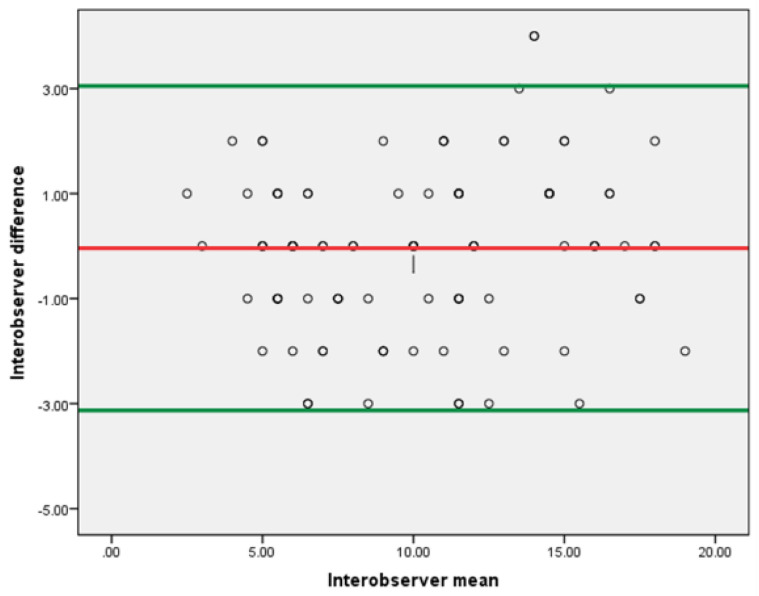
Bland–Altman plot. The Bland–Altman plot shows agreement between the two independent radiologists. The green lines indicate the 95% limits of agreement.

**Table 1 diagnostics-12-01529-t001:** Demographic, clinical and laboratory characteristics of the study population (n = 100); difference between survivor and non-survivor groups.

Variable Mean ± SD, n (%), Median (Min-Max)	Total (n = 100)	Non-Survivor Group (n = 60)	Survivor Group (n = 40)	P
* Demographic data *				
Age (years)	60±15	62.38±12.64	58.68±17.92	0.261
Gender				0.060
Male	54	37 (61.7)	17 (42.5)	
Female	46	23 (38.3)	23 (57.5)	
* Type of renal impairment *				**0.033**
Acute	53	37 (61.7)	16 (40)	
Chronic	47	23 (38.3)	24 (60)	
* Associated comorbidities *				
Diabetes Miletus	16	10 (16.7)	6 (15)	0.824
Hypertension	23	13 (21.7)	10 (25)	0.698
Ischemic heart disease	6	2 (3.3)	4 (10)	0.214
liver cirrhosis	10	9 (15)	1 (2.5)	**0.047**
Stroke	2	2 (3.3)	0	0.515
Atrial fibrillation	2	2 (3.3)	0	0.515
Systemic lupus erythematosus	2	2 (3.3)	0	0.515
Chronic anemia	16	9 (15)	7 (17.5)	0.738
Bronchial asthma	1	0	1 (2.5)	0.400
* Clinical presentation of COVID-19 infection *				
Fever	75	41 (68.3)	34 (85)	0.059
Dyspnea	86	51 (85)	35 (87.5)	0.724
Cough	65	39 (65)	26 (65)	1
Vomiting	1	0	1 (2.5)	0.400
Acute confusional state	6	5 (8.3)	1 (2.5)	0.397
* Vital data *				
Temperature (°C)	38.34 ± 0.93	38.56 ± 0.93	38.01 ± 0.83	** 0.003 **
Systolic blood pressure (mmHg)	128.83 ± 26.42	123.67 ± 27.37	136.57 ± 23.15	** 0.016 **
Diastolic blood pressure (mmHg)	79.08 ± 13.67	76.23 ± 14.01	83.36 ± 12.07	** 0.010 **
Respiratory rate (/min)	29 (14–50)	30 (14–50)	25.66 (15–34)	** 0.000 **
* Laboratory findings *				
White blood cells (× 10^9^/L)	9.45 (2.10–34.60)	9.65 (2.10–34.60)	9.35 (4.20–24.26)	0.992
Neutrophils (× 10^9^/L)	8 (1.38–31)	8.76 (1.38–31)	7.05 (2.2–20)	0.595
Lymphocytes (× 10^9^/L)	0.99 (0.2–5)	0.91 (0.2–2.7)	1.2 (0.4–5)	0.067
Platelets (× 10^9^/L)	182 (26–617)	165 (26–617)	235 (45–491)	**0.014**
Creatinine (mg/dL)	5.19 (2–21.76)	5.09 (2–21.76)	6 (2–21.07)	0.964
eGFR (mL/min/1.73 m^2^)	10.2 (1.79–43.37)	10.81 (1.79–43.37)	8.79 (1.87–28.45)	0.744
ALT (U/L)	26.5 (9–1004)	35.5 (11–1004)	22 (9–447)	**0.030**
AST (U/L)	37 (11–1897)	49 (15–1897)	27 (11–512)	**0.007**
Bilirubin (mg/dL)	0.7 (0.17–6.17)	0.8 (0.32–6.17)	0.56 (0.17–3)	**0.012**
Albumin (g/dL)	3.14 (1.80–4.47)	3 (1.8–4.2)	3.3 (1.9–4.47)	**0.010**
LDH (U/L)	632 (101.01–2636)	744 (150–2300)	440 (101–2636)	**0.001**
ESR (mm/hr)	60.26 (2–138)	59.5 (2–138)	62.29 (18–125)	0.984
D dimer (ng/mL)	709.03 (102.59–2237.94)	849.11 (135.11–2237.94)	594.45 (102.59–2194.59)	0.082
CRP (mg/L)	96 (3–366)	96.43 (12–366)	54.15 (3–234)	**0.003**
*Arterial blood gases*				
PH	*7.31* ± 0.099	7.31 ± 0.101	7.31 ± 0.09	0.76
PCO_2_ (mmHg)	33.01 ± 11.84	33.06 ± 11.97	32.94 ± 11.84	0.864
HCO_3_ (mEq/L)	19.4 (8.40–37.20)	19.9 (8.4–28.6)	18.95 (11.7–37.2)	0.861
Sodium (mEq/L)	133.95 ± 12.61	131.23 ± 10.30	135.77 ± 13.71	* 0.077 *
Potassium (mmol/L)	4.22 ± 1.09	4.21 ± 1.08	4.25 ± 1.12	* 0.853 *
SO_2_ (%)	* 91 (60–98) *	88 (60–97)	94 (75–98)	** * 0.001 * **
* Hospitalization *	94	58 (96.7)	36 (90)	0.169
Total duration of hospital stay (days)	6 (1–19)	6.5 (1–17)	6 (2–19)	0.628
ICU admission	83	57 (95)	26 (65)	**0.000**
Need for dialysis	49	27 (45)	22 (55)	0.327
*Need for O_2_ supply*	90	57 (95)	33 (82.5)	**0.005**
Invasive mechanical ventilation	26	23 (38.3)	3 (7.5)	** <0.001 **

ALT: alanine aminotransferase; AST: aspartate transaminase; CRP: C-reactive protein; eGFR: estimated glomerular filtration rate; ESR: erythrocyte sedimentation rate; ICU: intensive care unit; LDH: lactate dehydrogenase.

**Table 2 diagnostics-12-01529-t002:** Characteristics of HRCT chest, main patterns, and features in patients with COVID-19 infection, and differences between survivor and non-survivor groups.

Variable n (%), Median (Min-Max)	Total (n = 100) n	Non-Survivor Group (n = 60) n (%)	Survivor Group (n = 40) n (%)	P
*CT chest findings*				
Ground-glass opacities	100	60 (100)	40 (100)	-
Consolidation	71	54 (90)	17 (42.5)	**<0.001**
Crazy paving pattern	17	17 (28.3)	0	**<0.001**
Pleural effusion	49	34 (56.7)	15 (37.5)	0.060
Pulmonary nodules	7	6 (10)	1 (2.5)	0.238
Bronchial dilatation	1	1 (1.7)	0	1
Mediastinal lymphadenopathy	3	1 (1.7)	2 (5)	0.562
Pericardial effusion	4	2 (3.3)	2 (5)	1
*Predominant pattern*				
Ground-glass opacities	64	26 (43.3)	38 (95)	**<0.001**
Consolidation	34	32 (53.3)	2 (5)	
Crazy paving pattern	2	2 (3.3)	0	
*Laterality*				
Unilateral	2	0	2 (5)	0.158
Bilateral	98	60 (100)	38 (95)	
*CT chest total severity score*	10 (3–19)	12 (6–19)	6 (3–14)	**<0.001**

**Table 3 diagnostics-12-01529-t003:** Univariate logistic regression analysis of factors associated with mortality.

Variable	OR	95% CI	P
* Demographic data *			
Age	1.017	(0.990–1.044)	0.227
Gender			
Female	Ref		
Male	2.176	0.964–4.916	0.061
* Clinical data *			
Systolic blood pressure	0.981	0.965–0.997	**0.019**
Respiratory rate	1.149	1.057–1.248	**0.001**
* Laboratory finding *			
Bilirubin	2.532	1.099–5.836	**0.029**
LDH	1.001	1.000–1.003	**0.018**
CRP	1.010	1.002–1.017	**0.012**
ICU admission	0.033	0.004–0.261	**0.001**
* Need for O_2_ supply *	0.083	0.010–0.702	0.022
Simple face mask	0.474	0.100–2.241	0.346
Reservoir mask	1.588	0.454–5.558	0.469
Noninvasive ventilation	7.800	0.957–63.55	0.055
Invasive mechanical ventilation	7.667	2.118–27.755	**0.002**
* CT chest findings *			
Consolidation	0.082	0.029–0.235	**<0.001**
Pleural effusion	2.179	0.961–4.943	0.062
Pulmonary nodules	4.333	0.501–37.451	0.183
Mediastinal lymphadenopathy	0.322	0.028–3.676	0.362
Pericardial effusion	0.655	0.088–4.852	0.679
CT chest total severity score	2.082	1.579–2.745	**<0.001**

**Table 4 diagnostics-12-01529-t004:** Multivariate regression analysis to determine factors associated with mortality.

Independent Variable	Exp(B)	P
Constant	0.000	0.012
SBP	0.977	0.244
RR	1.085	0.327
Bilirubin	1.894	0.336
LDH	0.999	0.400
CRP	1.020	**0.046**
ICU	34.484	**0.040**
TSS	2.485	**<0.001**

## Data Availability

The datasets generated during and/or analyzed during the current study are available from the corresponding author on reasonable request.

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
