# Peer review of "Chest CT Total Severity Score on Admission to Predict In-Hospital Mortality in COVID-19 Patients with Acute and Chronic Renal Impairment"

_diagnostics, 2022, doi:10.3390/diagnostics12071529_

Round 1
Reviewer 1 Report
The manuscript has grammar, syntax and spacing errors.
Including GFR, severity of renal impairment and need for dialysis treatment in the analysis would make the results more clearer.
Did you do the multivariate regression analysis too?
Author Response
Reply to the reviewers
Dear respected editor and reviewers,
We wish to thank you all for your constructive comments on this manuscript. Your comments provided valuable insights to refine its contents and analysis. In this document, we try to address the issues raised as best as possible
Reviewer 1
The manuscript has grammar, syntax and spacing errors.
Reply The manuscript was edited, and grammar, syntax and spacing errors have been corrected.
Including GFR, severity of renal impairment and need for dialysis treatment in the analysis would make the results clearer.
Reply the GFR and need for dialysis were added to the results section at table 1.
Did you do the multivariate regression analysis too?
Reply Multivariate regression analysis was added to the results at table 4 .and added to the statistical analysis and results section.
If the reviewers are not satisfied, we can consider including more literature and give more details of our study. If the language is still not clear enough, we would be grateful if in the next round of revisions, you point to us the specific sentences we should improve
Once again, we thank you for the time you put in reviewing our paper and look forward to meeting your expectations. Since your inputs have been precious, in the eventuality of a publication, we would like to acknowledge your contribution explicitly.
Reviewer 2 Report
Coronavirus infection has become a worldwide problem, with millions of infected people and thousands of deaths. Data from the Italian Health Institute indicate that 26% of the coronavirus disease 19 (COVID-19) patients had 1 disease, 26% had 2 diseases, and 47% had 3 or more diseases, and only 1% of the patients had no other disease. The most common chronic pre-existing illnesses were arterial hypertension (76%), ischemic heart disease (37%), atrial fibrillation (26%), and active cancer within the previous 5 years (19%). All these comorbidities play a pivotal role in the augmented mortality among these patients, and renal impairment has been demonstrated to be a significant risk factor for COVID-19 mortality too.
The authors should be congratulated for the great work done on this paper.
It aims to identify the predictors of in-hospital mortality of COVID-19 patients with acute renal impairment (ARI) or chronic kidney disease (CKD) and evaluate the performance and inter-reader concordance of chest CT total severity score (TSS).
The authors performed a single-center retrospective study conducted on a cohort of 100 symptomatic patients with COVID-19 infection and renal impairment at the time of admission.
The manuscript is well-written and easily readable, tables and graphics are clear, but it is lacking in several points that would add value to the entire manuscript:
These two recent studies (https://www.ncbi.nlm.nih.gov/pmc/articles/PMC7360489/ and https://doi.org/10.1089/lap.2020.0251) report much information on COVID-19 disease that could be very interesting to include in the introduction of this paper to enhance its scientific appeal.
Author Response
Reply to the reviewers
Dear respected editor and reviewers,
We wish to thank you all for your constructive comments on this manuscript. Your comments provided valuable insights to refine its contents and analysis. In this document, we try to address the issues raised as best as possible
Reviewer 2
Coronavirus infection has become a worldwide problem, with millions of infected people and thousands of deaths. Data from the Italian Health Institute indicate that 26% of the coronavirus disease 19 (COVID-19) patients had 1 disease, 26% had 2 diseases, and 47% had 3 or more diseases, and only 1% of the patients had no other disease. The most common chronic pre-existing illnesses were arterial hypertension (76%), ischemic heart disease (37%), atrial fibrillation (26%), and active cancer within the previous 5 years (19%). All these comorbidities play a pivotal role in the augmented mortality among these patients, and renal impairment has been demonstrated to be a significant risk factor for COVID-19 mortality too.
The authors should be congratulated for the great work done on this paper.
It aims to identify the predictors of in-hospital mortality of COVID-19 patients with acute renal impairment (ARI) or chronic kidney disease (CKD) and evaluate the performance and inter-reader concordance of chest CT total severity score (TSS).
The authors performed a single-center retrospective study conducted on a cohort of 100 symptomatic patients with COVID-19 infection and renal impairment at the time of admission.
Reply Thank you very much for your encouraging words and thoughtful comments.
The manuscript is well-written and easily readable, tables and graphics are clear, but it is lacking in several points that would add value to the entire manuscript:
These two recent studies (https://www.ncbi.nlm.nih.gov/pmc/articles/PMC7360489/
Reply This study was referenced at the third paragraph of the introduction section
and https://doi.org/10.1089/lap.2020.0251) report much information on COVID-19 disease that could be very interesting to include in the introduction of this paper to enhance its scientific appeal.
Reply This study was referenced at the third paragraph of the introduction section
If the reviewers are not satisfied, we can consider including more literature and give more details of our study. If the language is still not clear enough, we would be grateful if in the next round of revisions, you point to us the specific sentences we should improve
Once again, we thank you for the time you put in reviewing our paper and look forward to meeting your expectations. Since your inputs have been precious, in the eventuality of a publication, we would like to acknowledge your contribution explicitly.
Round 2
Reviewer 2 Report
Authors answered all comments and suggestions.